# An Industrial IoT-Based Blockchain-Enabled Secure Searchable Encryption Approach for Healthcare Systems Using Neural Network

**DOI:** 10.3390/s22020572

**Published:** 2022-01-12

**Authors:** Aitizaz Ali, Mohammed Amin Almaiah, Fahima Hajjej, Muhammad Fermi Pasha, Ong Huey Fang, Rahim Khan, Jason Teo, Muhammad Zakarya

**Affiliations:** 1School of Information Technology, Monash University, Jalan Lagoon Selatan, Bandar Sunway, Subang Jaya 47500, Malaysia; aitizaz.ali@monash.edu (A.A.); muhammad.fermipasha@monash.edu (M.F.P.); ong.hueyfang@monash.edu (O.H.F.); 2Department of Computer Networks and Communications, College of Computer Science and Information Technology, King Faisal University, Al-Ahsa 31982, Saudi Arabia; malmaiah@kfu.edu.sa; 3Department of Information Systems, College of Computer and Information Sciences, Princess Nourah bint Abdulrahman University, P.O. Box 84428, Riyadh 11671, Saudi Arabia; fshajjej@pnu.edu.sa; 4Faculty of Computing and Informatics, University Malasia Sabah, Jalan UMS, Kota Kinabalu 88400, Malaysia; jtwteo@ums.edu.my; 5Department of Computer Science, Abdul Wali Khan University, Mardan 23200, Pakistan; mz@awkum.edu.pk

**Keywords:** homomorphic encryption, blockchain, security, healthcare system, smart contracts, privacy, performance

## Abstract

The IoT refers to the interconnection of things to the physical network that is embedded with software, sensors, and other devices to exchange information from one device to the other. The interconnection of devices means there is the possibility of challenges such as security, trustworthiness, reliability, confidentiality, and so on. To address these issues, we have proposed a novel group theory (GT)-based binary spring search (BSS) algorithm which consists of a hybrid deep neural network approach. The proposed approach effectively detects the intrusion within the IoT network. Initially, the privacy-preserving technology was implemented using a blockchain-based methodology. Security of patient health records (PHR) is the most critical aspect of cryptography over the Internet due to its value and importance, preferably in the Internet of Medical Things (IoMT). Search keywords access mechanism is one of the typical approaches used to access PHR from a database, but it is susceptible to various security vulnerabilities. Although blockchain-enabled healthcare systems provide security, it may lead to some loopholes in the existing state of the art. In literature, blockchain-enabled frameworks have been presented to resolve those issues. However, these methods have primarily focused on data storage and blockchain is used as a database. In this paper, blockchain as a distributed database is proposed with a homomorphic encryption technique to ensure a secure search and keywords-based access to the database. Additionally, the proposed approach provides a secure key revocation mechanism and updates various policies accordingly. As a result, a secure patient healthcare data access scheme is devised, which integrates blockchain and trust chain to fulfill the efficiency and security issues in the current schemes for sharing both types of digital healthcare data. Hence, our proposed approach provides more security, efficiency, and transparency with cost-effectiveness. We performed our simulations based on the blockchain-based tool Hyperledger Fabric and OrigionLab for analysis and evaluation. We compared our proposed results with the benchmark models, respectively. Our comparative analysis justifies that our proposed framework provides better security and searchable mechanism for the healthcare system.

## 1. Introduction

Data has been the center of all innovations in the technology industry. This has encouraged various organizations and vendors to implement technologies that allow interconnectivity to establish communications with different services. One of the main technologies that hassupported this movement is blockchain. Blockchain has been utilized to decentralize communications between different clients while maintaining anonymity and immutability in a trustless environment with no central authority. Blockchain enthusiasts have been proposing blockchain for different types of services and solutions. One of the proposed solutions for adopting blockchain capabilities is the paradigm of Internet of Things (IoT). Despite blockchain’s resilience and tamperproof capabilities, there have been some privacy and trust concerns raised due to its transparency. This research paper explores the privacy issues associated with the use of blockchains in Industrial Internet of Things (IIoT) solutions. Specifically, this paper explores the use of Ethereum blockchain in a proof of concept (PoC) environment while we perform threat modeling of external and internal threats as part of our research investigations. The results and outcomes from the experiments performed show clear issues with the privacy of the transactions occurring between the nodes in the blockchain, and present serious security risks to critical IIoT environments [1]. Encrypting sensitive information is the primary and essential strategy in cryptography when it comes to patient history details. The digital healthcare system is considered as the platform for transferring and receiving patient health records [2]. However, the existing healthcare systems lack security techniques, as most of them lack proper access control and encryption mechanisms. The distribution of healthcare data to authorized users is the critical requirement of efficient healthcare. More importantly, blockchain provides a peer-to-peer and decentralized network system. In general, blockchain can be classified into three different categories, namely, private, public, and consortium blockchain. It is a permission- and consortium-managed blockchain, which means all peers are known in the network. It provides trust and security to all the parties involved. Hyperledger Fabric is not domain-specific, and it supports Java, Go, Node.js, etc., for creating contracts and networks applications [3]. Several searchable encryption (SE) methods exist to provide a solution to the problems mentioned above, but they are not as efficient regarding flexibility and anonymity. SE can be categorized into different types based on several parameters such as single-write (SW), multiple-write (MW), single-read (SR) and multiple-read (MR) strategies. However, all searchable encryption approaches are inefficient when deploying to the cloud or server-based architecture systems. One of the most promising and secure approaches to solve these issues is secure, searchable encryption (SSE), which enables the users to encrypt the data at their side without the involvement of a third party. Secure, searchable encryption can be divided into two groups, named asymmetric SSE and symmetric SSE. Our proposed extended secure searchable encryption (ESSE) is based on the motivation of Cash et al. [4]. Cash et al. proposed the idea of an obvious cross tag (OXT) searchable mechanism. The idea of OXT is to distribute all master keys among the users to take more advantage of the protocol. The problem with OXT is the key loss or collusion attack, which make it more prone to vulnerabilities. Our proposed approach is more resilient to active collusion attacks, and key-loss situations [5]. Moreover, our proposed method can be applied to different platforms such as social media, fog computing, and other IoT-based applications [6]. In the Internet of Things, data is gathered via wireless networks and numerous physical sensors and analyzed in real time. The data gathered and analyzed is used to operate the actuators [7]. As a result, in the holistic development of IoT infrastructure in smart cities, critical factors such as centralization, scalability, trust, privacy, and security must be ensured. IoT systems, on the other hand, are heterogeneous and highly distributed in nature, and thus differ from traditional systems. Because of the unique characteristics of IoT, such as trust, privacy, IoT security maintenance, battery life, network bandwidth, processing capabilities, and memory capacity, the design of sustainable smart cities has proven problematic. However, the interconnectedness of numerous IoT sensors in smart networks creates a slew of potential aimed attacks [8]. Cyber and physical attacks are two kinds of attacks involved in smart cities. Sleep denial Attack, side channel attack, permanent denial of service, fake node injection, radio frequency jamming, and malicious code injection are examples of these assaults [9]. In a cyberattack, the attacker attempts to inject malicious software or malware into network components in order to gain unauthorized access to them. Ransomware, distributed denial-of-service (DDoS), and man-in-the-middle attacks (MITM) are all examples of these types of attacks [5]. In this study, we propose a novel framework for the privacy-preserving model in IoT. The major point of this research is as follows: Our proposed algorithms converts data into a new reduced structure for attack prevention and poisoning. The proposed model detects intrusion and nonintrusion data.

In the trustworthiness module, we created an address-based blockchain reputation system. In this research paper, we propose extended multi-users extended secure, searchable encryption, which supports the participants to query securely against desired keyword search in the distributed ledger. The patient encrypts the data at the beginning and uploads it to the blockchain. Our research method provides facility to the data owner once the data owner completes the encryption; it will not be necessary to be involved in other processes until they need policy revocation or deletion. The rest of the article is summarized as below: Section 2 describes the related work, followed by a description of the proposed work in Section 3. Section 4 discusses the experimental investigations. Finally, the paper is concluded in Section 5.

### 1.1. Motivation

The existing access control system only relies on identity-based, role-based, or attribute-based methods. Through analysis and comparison, it is observed that ABE is the optimal access control model among existing access models [7]. However, the public-key encryption does not fulfill the security requirements for attribute-based encryption. In our proposed approach, we use attribute-based signature (ABS) because it provides unforgettability and anonymity of the signer [8]. One of the main motivating problems of our proposed scheme is giving security to the subject, object, and personal health record (PHR) with both vital data confidentiality, and flexible fine-grained user anonymity access control without imposing an additional cost on them. We propose a novel data-sharing protocol by combining and exploiting two of the latest attribute-based cryptographic techniques to achieve our goals. They are attribute-based encryption (ABE) and attribute-based signature (ABS) with trust-based access control (TBAC) model using distributed ledger fabric [9]. Furthermore, we also provide a detailed comparison of our proposed scheme with several of the latest existing schemes. The development of urban areas environment, as well as and communication technology (ICT) industries, refer to the term ‘‘smart city’’. There is a global trend toward smart cities as global urbanization continues to expand, with the overall population anticipated to double by 2050 [10]. In order to accommodate the expanding population, cities require infrastructure and amenities to address transportation and environmental concerns. As a result, smart cities have emerged as a viable solution to the aforementioned issues. The rapid development of low-cost devices, including radio-frequency identification (RFID) devices, sensors, and actuators, are combined with the Internet of Things (IoT)-oriented infrastructure and wireless communication technology. For smart city applications, one of the major enabling technologies is the IoT that refers to the use of Internet technology to connect computer devices. In the Internet of Things, data is gathered via wireless networks and numerous physical sensors and analyzed in real time. The data gathered and analyzed is used to operate the actuators [11]. As a result, in the holistic development of IoT infrastructure in smart cities, critical factors, such as centralization, scalability, trust, privacy, and security, must be ensured. IoT systems, on the other hand, are heterogeneous and highly distributed in nature, and thus differ from traditional systems. Because of the unique characteristics of IoT, such as trust, privacy, IoT security maintenance, battery life, network bandwidth, processing capabilities, and memory capacity, the design of sustainable smart cities has proven problematic. However, the interconnectedness of numerous IoT sensors in smart networks creates a slew of potential aimed attacks. This research paper discusses the privacy issues associated with the use of consortium blockchain in IIoT environments. The reason for choosing Hyperledger Fabric blockchain is due to it being one of the most popular blockchain ecosystems utilized for implementing decentralized applications (Dapps). A report was carried out in mid-2020 by Coin Telegraph which shows that the number of Hyperledger Fabric accounts is around four times the number of Bitcoin addresses, highlighting its popularity among blockchain enthusiasts.There are various implementations of blockchain that utilize different frameworks and protocols based on the application they serve. These include, but are not limited to, the well-known Bitcoin [12] and Hyperledger [13]. There are multiple blockchain use-cases covering various industries, including healthcare [14], energy utilities, and government sector [15]. The significant difference between our proposed scheme and the traditional one is the application of the computational trust value.

Our proposed approach will examine the parameters chosen, including user behavior, attributes, trust, unauthorized request, forbidden request, and specification range. Users will be divided into different categories based on the trust value: very low, low, unknown, moderate, high, and very high trusted users. A threshold value will be set if a user meets that threshold value and the policy; then, access will be granted [16].

### 1.2. Contributions

The major contributions of this paper are as follows:A detailed literature review of the state-of-the-art patient and participants detection based on encryption ad security algorithms.Novel cross-domain and access control policies are proposed using homomorphic encryption.We propose the idea and implementation of policies revocation, updates, delete and add using homomorphic encryption.We achieve optimum security and anonymous keyword search in the Hyperledger Fabric framework.Our proposed research method provides an alternative private key in case the key is lost.We achieve efficiency compared to the existing methods, as these methods exhibit more communication and encryption cost because they need to encrypt the data. Our proposed plans provide a more efficient solution to the users.

In this section, we discussed the study and the loopholes found in the previous research. The rest of the paper is organized as follows. We divide our literature review into two sections. First, we present literature on the current and earlier methods used for the patient health record system (PHR)—Section 2. The second part describes literature on the topic or literature on the access control model with weaknesses and strengths—Section 3. In Section 4, we discuss our proposed algorithms. Performance evaluation of the suggested algorithms is discussed in Section 5. Finally, Section 6 concludes this paper with directions for future research.

## 2. Literature Review

To access the patient health record, URLs of the PHR are stored on the blockchain. However, the exiting approaches based on blockchain for patient health record or clinical record do not provide efficient access control for data search over the blockchain. As most of the recent research shows, blockchain is not search-friendly; looking up a specific record would be very slow with data increase. Kim et al. [11] a novel method to manage blockchain-based human resource management. The method is based on the distributed ledger method (DL). The authors described the utilized privacy-preserving technique that is used to offer a transparent system while managing the human resource record. The public–private key pair was generated along with the organization ID, anonymity mapping, and hash [15]. The performances of the work were analyzed based on failure point identification, time, read–write latencies, and memory consumption. Meanwhile, it achieved better performance for all parameters except the time consumption. However, the execution is high. Kumar et al. [6] presented a novel deep blockchain-based trustworthy privacy-preserving secured structure (DBTPPS) to address the challenges such as privacy, trust, security, and centralization factors. The authors enclosed three modules: two-level privacy preservation module, trust management, and an anomaly detection module [16]. The first module was based on BC incorporated with autoencoder. The trustworthiness module composed a BC-based address reputation system. The last one was included in a deep neural network-based approach. The detection rate and accuracy of the presented work were 93.87 percent and 98.97 percent, respectively. However, they did not estimate the overall effectiveness of the proposed approach [17]. Le et al. stated a novel approach known as ant colony optimization (ACO) for privacy-preserving and for secure and reliable IoT data sharing; they adopted multi-kernel support vector machine along with elliptical curve cryptosystem (ECC). The protection and integrity were obtained by the blockchain. The experimental analysis depicted that the work attains better precision and recall and thus ensures security, privacy, confidentiality, and reliability. Nevertheless, the privacy of multiple components of encrypted datasets is difficult to attain. Qi et al. [18] presented a novel method known as blockchain-based federated learning (BFL) approach to preserve privacy for the traffic flow prediction. The authors also described that the method can be used for the enabling of reliability and decentralized ad secured federate learning without the inclusion of a centralized model coordinator. The work provided better privacy protection and circumvented the data poisoning attacks. However, the communication overhead is a little higher in this approach. Shala et al. [19] delineated a novel optimized trust model incorporated with a multilayer adaptive and trust-based weighting system. The authors presented numerical approaches to trust estimation clearly. The resiliency ad reliability of the method is maximum, however, the authors did not focus on the control–loop concept and their integration to achieve a decentralized IoT system. Wu et al. [20] stated a novel blockchain-based trust management mechanism (BBTM) to provide better trustworthiness of the sensor nodes. The authors described better trust estimation and checked the estimation process. This work achieved better trust accuracy, resilience, and convergence against the attacks. However, there is no possibility for real-time application.

In the traditional symmetric critical model, encryption is carried out using a symmetric key. The data owner divides data into some groups and then encrypts these groups using the symmetric key. Users who have private keys can decode the encrypted data. In this scheme, authorized users are listed in the ACL [21]. The major drawback of this scheme is that the number of keys grows linearly as the number of data groups increased. In addition, supposing any change occurs in the user and data owner relationship, it will affect other users in the ACL. Therefore, in summary, this scheme is not practical for use in different scenarios [22]. Finextra mentioned, in their 2021 expected trends for blockchain, that it is expected that the global blockchain market will expand to USD 39.7 billion by 2025 [23]. In 2019, Deloitte Global blockchain survey revealed that blockchain is undergoing a phase of industry expansion in sectors such as telecommunication and health beyond its initial main use within fintech applications [24]. In addition, it was forecasted by Gartner that blockchain will generate more than USD 3 trillion of annual business value by 2030 [25]. With these interesting reports and trends, there has been some interest in adopting blockchain in IoT-based solutions. This is due to the large market that IoT has in multiple sections in the past few years. A report produced by Business Insider states that by year 2027, the IoT market will reach annual growth of over USD 2.4 trillion  [26]. With this enormous growth, a huge interest in digitizing industrial assets was revealed by the Industry 4.0 initiative. Industry 4.0 is the fourth industrial revolution which consists of the following trends, including artificial intelligence (AI), advanced automation, and data analytics [27]. However, with this enormous growth, there are security risks that come into effect when this increased connectivity is enabled within critical infrastructures such as IIoT [28].

### State of the Art

One of the key benefits that blockchain aims to achieve is high integrity and availability through its transparency and decentralized nature. This prevents any possible tampering of data and maintains the integrity of the solution running on the blockchain. Multiple research papers have explored the privacy issues raised from the transparency of blockchain solutions. Authors of [29] have explored the idea of utilizing graph theoretic data mining techniques to visualize and construct a graphical view of a consortium blockchain using Hyperledger Fabric blockchain-based transactions [30]. This technique has raised issues related to the privacy of nodes and hyperledger accounts where typical transactional behavior can assist in identifying some of the active participants of the blockchain. In addition, this technique used by the authors aimed to extract the most active address and central address of the blockchain. This may allow an attack to target this main address and take control of this account [31]. In addition, the leakage of transactional privacy was discussed by authors in [32]. This also holds true for smart contracts as they are completely visible and may contain exploitable vulnerabilities that could place the blockchain and its participants at risk. While these risks were explored generally for blockchain, they have not been deeply analyzed in context to IIoT environments that are considered to be critical infrastructures [33]. By performing appropriate threat modeling of a consortium blockchain solution, our paper examines the following research questions:What are the threats that IIoT can face when blockchain is utilized in the integrated framework?How can blockchain transparency impact the exposure of IIoT environments to external threats?What are the implications of compromising blockchain nodes within IIoT environments?

## 3. Preliminary Data

This section includes what we already know about blockchain, trust, and the patient health record system. This section also describes the fundamentals of the preliminary data, research findings, and the importance of methodology [32].

### 3.1. Blockchain in Healthcare System Using Hyperledger System

With the use of blockchain technology, transparency and communication between patients and healthcare providers are also enhanced. An overview of blockchain technology and its working in the healthcare industry can be found in [33]. The figure below shows a traditional centralized technology that solely relies on a centralized server—as shown in Figure 7.

#### Blockchain Technology

The uses of blockchain in digital healthcare systems have an essential role in the present digital health industry [34]. Data distribution, redundancy, and fault tolerance are such features that are supported by blockchain. We propose a new access control method to achieve trust with secure access control using blockchain through this research. Our proposed framework bypasses dependencies on the CA and an SOP in the framework [35,36]. In our proposed framework, immutable technology is used to achieve system security. For performance evaluation, we used Hyperledger Calliper for the proposed system. We used different scenarios for our experiment through the variation of the size of a block, the creation time of a block, designed policy, and proposed method for evaluating such metrics [37]. These metrics contain delay, throughput, and PHR security to achieve optimized results [22] Through performance optimization, the proposed system will ultimately improve latency, security, and increase trust. In addition, our proposed research will prove the blockchain application and importance in the digital healthcare system in various aspects and justify that it can be the succeeding technology for substituting traditional health models [38]. In Figure 1, we explain the application of blockchain technology in various domains. The applications are growing with respect to time and advancement in technology [39].

The equations for several rounds and transactions for PHR is represented below:(1)$$\left[y_{i}f_{i}=f\left({(}_{i}^{n})\sum \left(xi,wi\right)\right)\right]$$
(2)$$\left[f_{x}=tanh\left(x\right)=\left[2/1+exp\left(-2x\right)\right]-\right]$$
(3)ETx(k,d)=ETx−Elec(k)+ETx−amp(k,d)
(4)ERx(k,d)=ERx−Elec(k)
(5)ERx(k)=Eelec×k
(6)$$P^{L}\left(f\right)\infty f^{k}$$
(7)$$P^{L}\left(f,d\right)=PLo+10nlog10d/do+X\alpha$$
(8)$$P^{L}o=10log10$$
(9)$$\left(4\Phi \times d\times f\right)c^{2}$$
where *n* is the number of neurons in the previous layer, and in the case of activation function, the most common one is the hyperbolic tangent. Still, the selection is made concerning expected output. The above equations are used for the classification of the users’ behavior and interaction with the system using neural network. We divided our datasets into two categories, i.e., the first is training dataset and the second is testing dataset. We used 30% of data as training and 70% as testing.

## 4. Proposed Secure Search Algorithm

We designed a novel secure, searchable algorithm that allows the users to encrypt at their own side and upload it to the distributed ledger. Through our proposed extended secure, searchable algorithm, users can anonymously search the keywords using blockchain users API. If the user loses the key, they can revoke the policy and request a new key. It protects against active collusion attacks. The list of various parameters and mathematical notation used in our proposed framework are shown in Table 1:

Algorithm 1 defines the attributes-based signature techniques. Our proposed access control is based on attributes and feature selection. If a user fulfills the criteria based on the required attributes, then he is given access, otherwise the access is denied. Our proposed access control mechanism uses hybrid neural network with attribute-based access control, which makes it flexible and more secure. Our proposed framework consists of four main participants, i.e., admin, doctor, patient, and lab technician. We propose delegation policies and algorithms for each node.   
**Algorithm 1:** Attribute Based Signing Algorithm
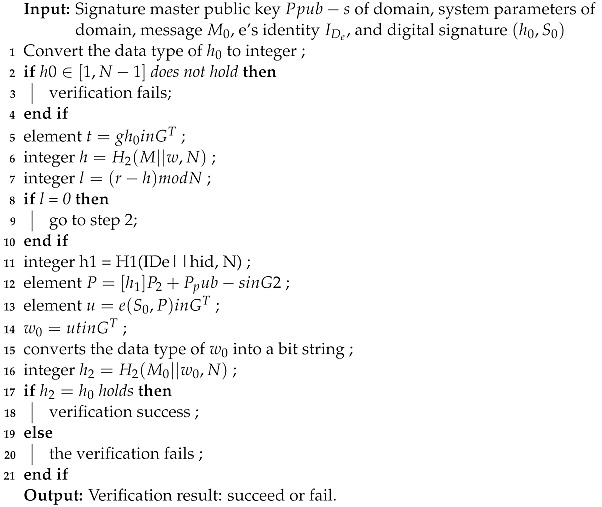


### 4.1. Proposed Access Control System for Framework

We propose a novel, secure access control system based on attributes based on our proposed framework and the proposed access control.

### 4.2. Proposed Algorithm

The proposed enhanced homomorphic encryption(EHE), as mentioned in Algorithm 2, consists of setup, initialization, update, and search steps. The setup step provides the configuring to the algorithm, where initialization provisions initialize the parameters. This section develops the binary description of the spring search (BSS) algorithm. The binary number, such as 1 and 0, describes the binary version of SSA. Because the search space is discrete, each variable on the axis must be represented by the appropriate number of binary values. Because there are only two integers in the binary version, i.e., one and zero, the idea of displacement is defined as altering the status from zero to one or one to zero [16]. A probability function is used to implement the idea of displacement in the binary version. The new location of each member in each dimension of the issue may change or remain intact depending on the value of this probability function. The probability I D Y is I D DY that becomes one or zero in the BSS algorithm. In both binary and real versions, the steps are updated similar to the number of displacement of peer members of the population. The constant values of the spring calculate the constant values of spring [17]. The population in which the difference among the two versions is updated, where 0 and 1 are the probability function. The below equation calculates the position of each member dimension. The probability of each member of the population varies its position based on the above equation. In dimension D, the higher the object i, indicates the probability moving with the higher value of i = $D_{Y}$. The normal distribution tends to the interval 0 to 1 based on the random number. The standard functions are considered in order to describe how to seek the optimal solution.

### 4.3. Hybrid Neural Network Algorithm

The steps involved in combining hybrid deep neural network (HDNN) algorithms for optimization are as follows: Initialization of neural network parameters with a maximum number of iterations. Determine the constraints of the optimal solution. Evaluation of fitness functions and the constraints that these functions impose. The multi-balanced neural network algorithm selects the optimal solution at two levels. The group theory optimization algorithm selects the best transaction time. The binary search algorithm algorithm selects the best route within the blockchain.

### 4.4. Revocation Policy for Proposed Framework

Due to the collusion attack, our system will monitor the user’s behavior and interaction with the system. To remove the colluded node or user, we propose the revocation policy. The shared key in the blockchain access control policies is revoked, and a new share key will be created among the shareholder.

### 4.5. Update Policy and Proposed Algorithm

To implement the updated policy, we propose our novel algorithm, called update policy. In the case of the data, the owner has lost the private key, so the update algorithm can request a new private key.
**Algorithm 2:** Homomorphic Encryption
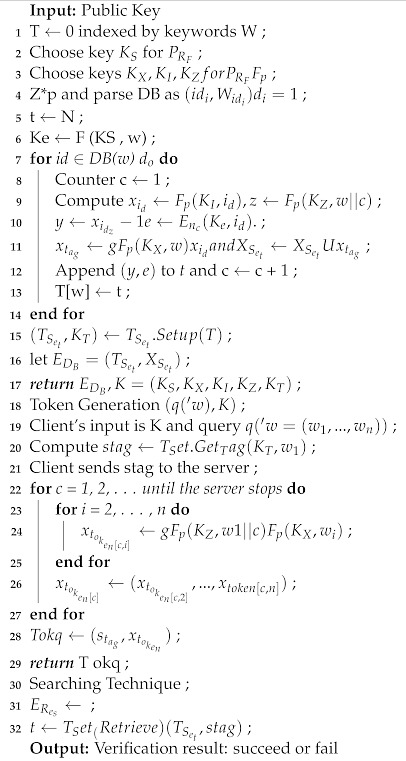


### 4.6. Proposed Methodology

We propose a blockchain-based access control and secure searchable encryption system to solve the challenges and issues highlighted in the literature in multisite clinical systems. It is used for keyword searching, storing, retrieving, and sharing personal healthcare data using homomorphic encryption. We model our approach on Hyperledger Fabric and use homomorphic encryption for security and secure search. Figure 2 represents our proposed neural network for proposed framework. The proposed NN model consist of several layers, and each layer carries specific information. We divided our dataset of IoT into two sections, i.e., training dataset and test dataset. Our proposed algorithms are embedded in smart contracts for blockchain technology, and we have described all our novel algorithms function in detail. The parameters and the notations that are contained in the blockchain are illustrated in tabular form. This section describes the design of our proposed system: setting up the network, installing private channels, and writing channel-specific intelligent contracts. Figure 2 and Figure 3 show the function of the blockchain-based system and access control decision system, alternatively [40,41].

### 4.7. Proposed Data-Sharing Scheme

In Figure 3 we explain the working of our proposed framework for data sharing using blockchain technology. From Figure 3, it is very obvious that the proposed model consists of federated blockchain system which interconnects a smart city, healthcare system, and financial institution using a neural network system. In Figure 4, we provide the comparative analysis of keyword search results using homomorphic encryption and the confirmation time in seconds. From the results and comparative analysis, it is evident that our proposed framework performs better than the benchmark models, respectively. Moreover, the confirmation time is considered significantly less in the case of our proposed framework, so ultimately, the throughput of our proposed framework is more, compared to the benchmark models.

**Figure 3 sensors-22-00572-f003:**
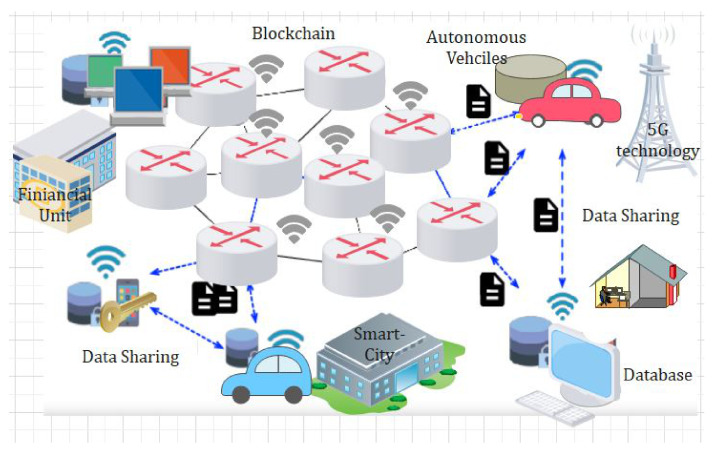
Proposed data-sharing scheme.

**Figure 4 sensors-22-00572-f004:**
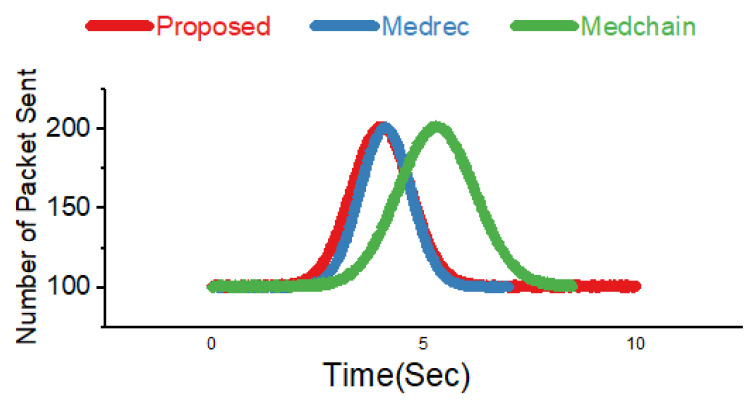
Comparative analysis of different blockchain-based domains.

### 4.8. Security Analysis

As IIoT environments are commonly purposed for serving critical infrastructures that are of a high risk, the main access control security principals must apply at all times. This includes, but is not limited to, the principle of least privilege and role-based access control policies. Ethereum blockchain has been mostly designed to allow connectivity between all the blockchain nodes to increase the verifiability, and hence integrity, of the solution. To analyze the interactivity and flexibility of an Hyperledger Fabric solution, we implemented a consortium blockchain topology consisting of three node peers (N1, N2, N3) running on virtual machines (VMs), shown in Figure 5. Each VM was running Ubuntu OS, which is the light version of Ubuntu Linux distribution with 1 vCPU and 2 GB RAM. In this PoC, we demonstrate the transparency implemented in consortium blockchain using the above deployed topology. Table 2 displays the three randomly generated account addresses that will be used by the three deployed nodes in this topology. We started this proof of concept (PoC) by connecting all the three nodes together in the same chain. Figure 2 is a screenshot of the blockchain operations occurring live on node N3.

Cyber and physical attacks are two kinds of attacks involved in smart cities. Sleep denial attack, side channel attack, permanent denial of service, fake node injection, radio frequency jamming, and malicious code injection are examples of these assaults [42]. In a cyberattack, the attacker attempts to inject malicious or malware software into network components in order to gain unauthorized access to them. Ransomware, distributed denial-of-service (DDoS), and man-in-the-middle attacks (MITM) are all examples of these types of attacks [43]. In this study, we propose a novel framework for the privacy-preserving model in IoT. The major point of this research is as follows: In the trustworthiness module, we create an address-based blockchain reputation system, and the MICA converts data into a new reduced structure for attack prevention and poisoning. The proposed HDNN model detects intrusion and nonintrusion data. The rest of the article is summarized as below: Section 2 describes the related work, followed by a description of the proposed work in Section 3. Section 4 discusses the experimental investigations. Finally, the paper is concluded in Section 5.

### 4.9. Intrusion Detection Module

One type of feedforward neural network is the convolutional neural network (CNN), which has less network complexity. CNN is applied in several fields such as fault diagnosis, natural language processing, and computer vision, and is explained through Figure 3. In this section, we propose a hybrid deep neural network (HDNN) for accurate intrusion detection, which is made up of two branch networks. The model’s input is made up of 51-dimensional data that is divided into two portions. When one is an LSTM neural network and the other is a residual CNN-LSTM neural network, one branch is residual CNN-LSTM. The final results are provided by four layers of full-connection layer networks concatenating the parallel network outputs. The residual CNN-LSTM neural network consists of three layers of the LSTM network and two levels of CNN. As a result, three layers of the LSTM network are available in the other branch’s LSTM neural network [44].

### 4.10. Security Threat Model

In this section, we perform threat modeling to identify the different threats and attack techniques that can be used against this PoC environment. We used one of the most known threat models, called STRIDE (spoofing, tampering, repudiation, information disclosure, denial of service (DoS), and elevation of privilege). STRIDE focuses on identifying the following threats and their affected security property, as shown in Table 3 [45]. STRIDE is generally used to categorize and identify threat vectors in threat modeling. It has been examined in the past to be used along with the industry-known MITRE ATT&CK framework to identify threats presented as tactics, techniques and procedures (TTPs). There are two types of threats which target different industries and sectors; internal threats and external threats. Internal threats exist within organizations’ trusted boundaries. Internal threat sources can be categorized as follows:Intended malicious insider: intent to affect the confidentiality, integrity, and/or availability of systems and data.Unintended innocent insider: person working for the organization making a human error in their day-to-day duties.Compromised insider: involves compromising an employee’s user account due to the lack of security awareness from sources such as phishing and trojans.External threats are generated from the exploitation of internal vulnerabilities to assist attackers to gain access to environments.

The attack sources can be categorized as follows:Malicious actor.Compromised supply chain.External insider threats.

**Table 3 sensors-22-00572-t003:** Performance analysis based on number of patients.

Number of People	FPR	FNR	FDR	ACC
100	0	0	0	1
200	0	0.022	0.025	0.96
300	0.002	0.029	0.035	0.87

Using the STRIDE model, there are a few external and internal threats that pose risks to this solution. This consists of the following: Internal threats External threat attacks

T1—A malicious insider can gain access to all the interactions occurring in a network through nodes and store them for malicious purposes [46].

T2—A node can be compromised by a malicious actor. This can expose all the blockchain and allow the actor to get the full transaction and blockchain history continuously.

T3—A malicious attacker can try to find some vulnerabilities in existing deployed smart contract.

T4—A malicious attacker may find some sensitive/secret information exposed in the transactions and/or the smart contracts such as private credentials [45,47,48].

## 5. Experimental Results

In our proposed research, Hyperledger Calliper is used as a tool for the blockchain network. It can support different Hyperledger frameworks, e.g., Fabric, Composer, Sawtooth, Iroha, etc. Moreover, we have implemented homomorphic encryption for our encryption and decryption to provide a secure, searchable encryption mechanism. In this proposed research, the Calliper tool plays an important role in the verification and execution of the system and various parameters. The parameters include latency, throughput, encryption and decryption time, and computational cost. In our experimental setup, the configuration parameters are modified as per assessment, such as block size, block time, endorsement policy, channel, keyword search, update policy, add a policy, delete policy, and revoke the policy. Our simulation setup configurations consist of the following specifications:

**Experiment 1:** We ran our first experiment up to 3050 rounds, and we evaluated our results based on the number of personal health records sent versus several rounds.

In Figure 4, we explain the number of transactions sent from one domain to another domain. It can be easily observed that the number of transactions means the number of patient health records (PHR) or electronic health records (EHR) sent per round. We ran our simulations for 5000 rounds and evaluated the number of patient health records sent. In addition, we carried out a comparative analysis with benchmark models, such as Medrec and Medblock.

**Experiment 2:** We performed our simulations for the number of rounds and the execution time to compare the proposed framework and the benchmark models.

In Figure 6, we describe the simulation results based on our proposed policies. We proposed access control policies for our proposed framework using homomorphic encryption and pseudorandom algorithms. The proposed method evaluates access control policies against several execution times and several access policies. In addition, we carried out experiments on policy revocation, policy creation, and add policy.

It can be observed that the authorization policy took less time than the authentication policy and delegation policy. Thus, these simulations in Figure 7 justify that our proposed access control policy provides more security and less computational cost. Figure 8 describes the simulation results of several users classified based on their interaction and behavior with the proposed framework.

**Figure 6 sensors-22-00572-f006:**
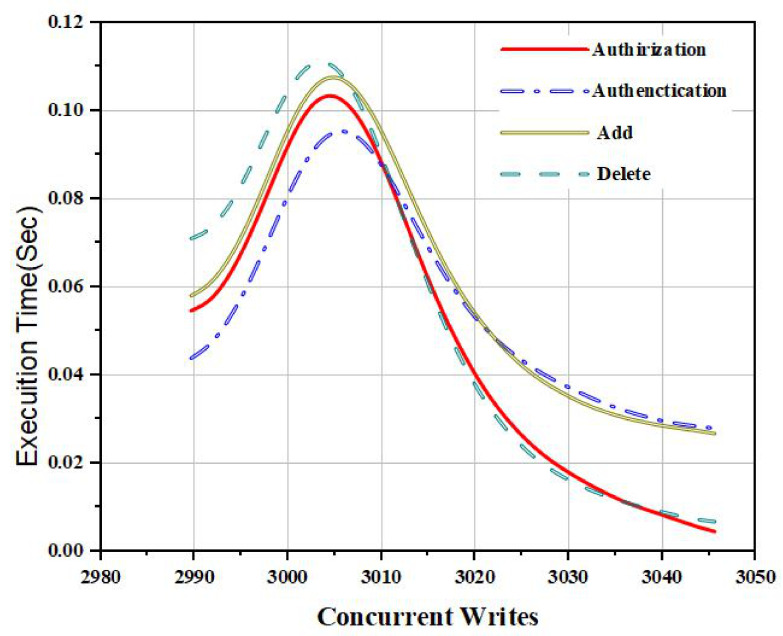
Comparative analysis of different domains based on homomorphic encryption and secure searchable.

**Figure 7 sensors-22-00572-f007:**
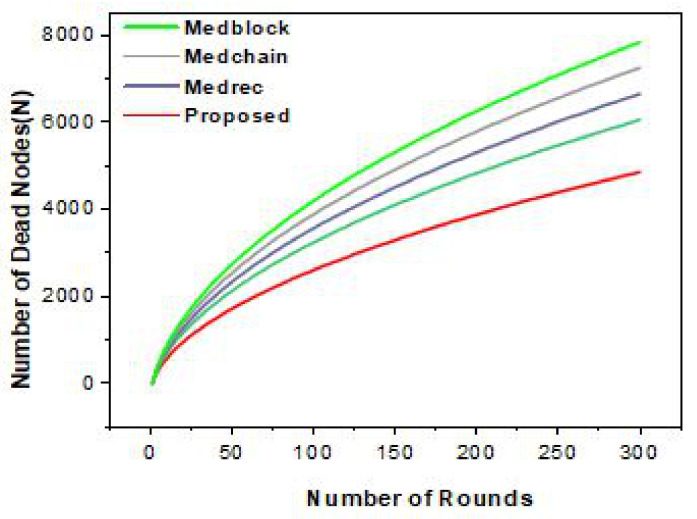
Comparative analysis of concurrent requests for the proposed policies.

Figure 4 highlights the simulation results for the number of rounds taken and the number of transactions sent per second. From the simulations, we can see that our proposed framework is much better than the benchmark models. Thus, we have achieved more efficiency as compared to the benchmark models.

In Figure 6, we achieve the throughput and efficiency using the Hyperledger Fabric tool. Through our proposed framework, we used the optimum block height to achieve the maximum throughput. The gas is the space or the unit during the transactions used. We evaluated this experiment over several rounds as the input and the number of packets sent to the cluster as the output. From these simulations, it is evident that we achieved the maximum efficiency and throughput for the same dataset used in the literature, i.e., Medrec and Medblock.

In Figure 7, we provide the comparative analysis based on the number of transactions sent and the time for transactions. We designed a novel algorithm for the transaction of personal health records using blockchain technology. Through the PHR proposed algorithm, the user can encrypt the clinical and patient data and upload it to the distributed ledger. Our proposed algorithm eliminates the involvement of the blockchain manager.

**Figure 8 sensors-22-00572-f008:**
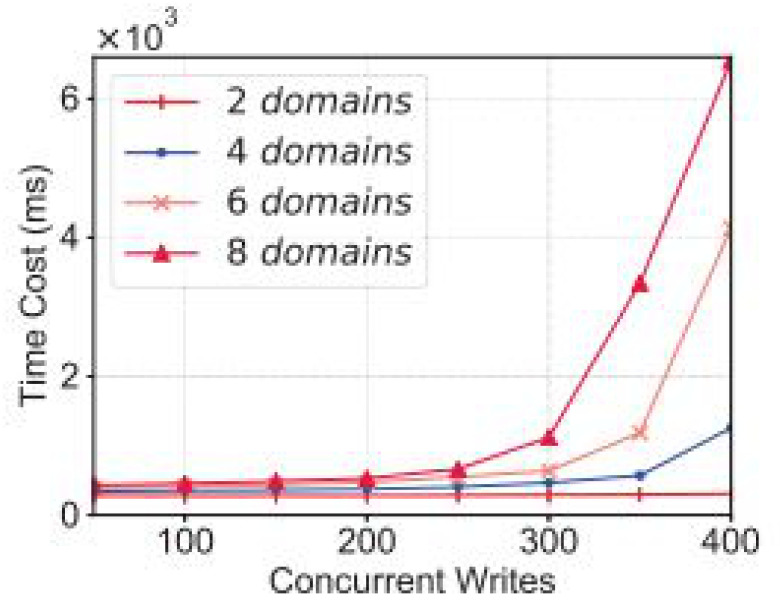
Performance comparison of the proposed framework and Medrec.

In Figure 9, we provide the comparative analysis of the proposed framework test accuracy and global epoch. The comparison was carried out based on the number of keyword searches and the confirmation time in seconds. The proposed algorithm for the keyword search is explained in our proposed framework.

## 6. Conclusions and Future Work

Blockchain has been one of the most hyped technologies in the past 5 years due to its popularity gained by its various cryptocurrencies. There have been multiple use cases that were implemented using Bitcoin, Ethereum, and other blockchain technologies. However, none of these use cases covered critical infrastructure with sensitive systems and data as their assets. While blockchains including Ethereum provide important anonymity, integrity, and auditability features for its users, there are important privacy and security risks that were discussed and presented in this paper related to their use in critical environments, such as IIoT environments. These privacy issues exist in other blockchains as one of their main design principles utilizes distribution of ledger. There are future improvements in the current roadmap of Ethereum 2.0 that address the privacy issues discussed in this paper. However, with all the additional security and privacy features, it is very important to analyze and study the performance of any blockchain framework prior to deploying it in latency-sensitive environments. We implemented the novel comprehensive approach of homomorphic encryption in the digital healthcare system using blockchain technology that provides a secure keyword search facility at the user’s end. Our research method supports immutability and tamper-resistance, and delivers secure data, which reduces security breaches to the healthcare data. Furthermore, our novel mechanism allows blockchain users to encrypt data at their side and upload to the distributed ledger for record purpose. Users can securely search the desired health-related data without decryption based on homomorphic SSE. Furthermore, it provides resistance to active cooperation and replays attacks due to the flexible policy revocation. Blockchain technology also supports distributed data, redundancy, and fault tolerance features for digital systems. In this proposed research, current challenges and problems in the literature faced by the digital healthcare industry will be solved. We propose a framework and algorithm that enables access control policy for users to achieve privacy and security for patient health data in the PHR system. The proposed method provides more independence to the users, and it supports flexibility and fine-grained keyword search. We have justified our proposed research algorithms and policies through simulations run on the Hyperledger Fabric tool. We used the Pycharm tool for data analysis. With our proposed method as the most up-to-date approach applied first on healthcare and blockchain technology, we have improved security and anonymity, compared to the benchmark models, such as Medrec, Medchain, and Medbichain, respectively.

## Figures and Tables

**Figure 1 sensors-22-00572-f001:**
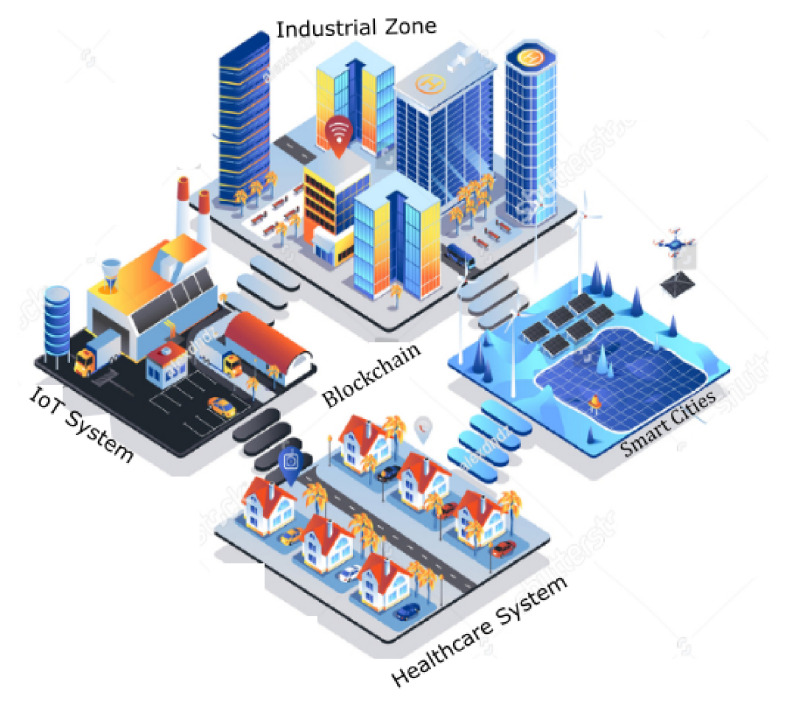
Applications of blockchain technology.

**Figure 2 sensors-22-00572-f002:**
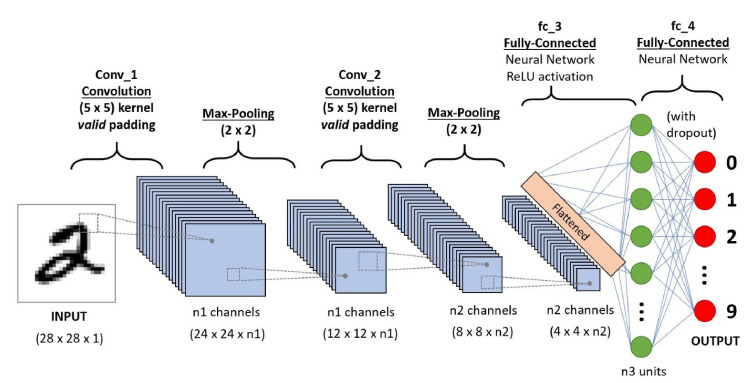
Performance comparison of the proposed framework and Medrec.

**Figure 5 sensors-22-00572-f005:**
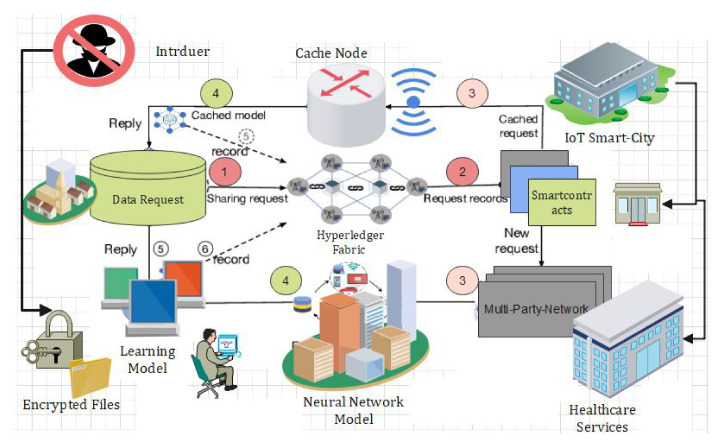
Performance comparison of the proposed framework and Medrec.

**Figure 9 sensors-22-00572-f009:**
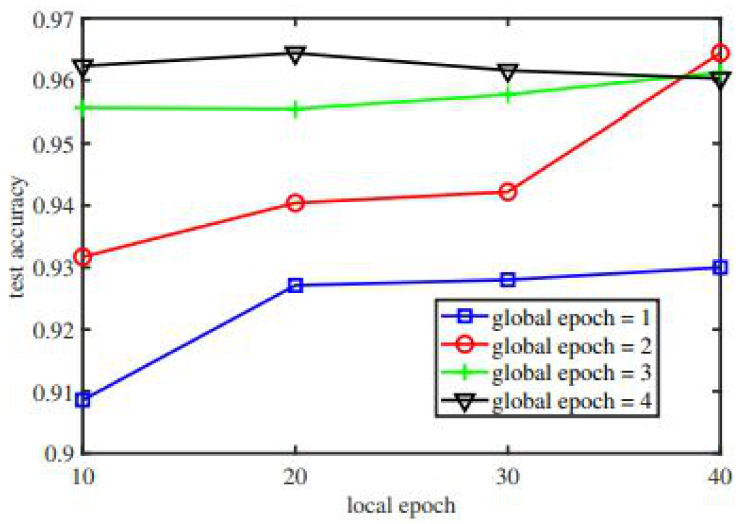
Performance comparison of the proposed framework and Medrec.

**Table 1 sensors-22-00572-t001:** List of parameters for our proposed algorithms.

S. No	Parameters	Details
1	BN	Blockchain network
2	CID	Clinician ID
3	LID	Lab ID
4	PHR	Patient health record
5	$R^{s}$	Ring signature
6	$U^{Name}$	Username
7	$P^{K}$	Private key
8	*r*	Integer
9	*N*	Number of nodes
10	*G*	Bilinear order group
11	$P^{1}$	Generator of additive group 1
12	$P^{2}$	Generator of additive group 2
13	id	Bilinear identifier
14	*H*	Homomorphic encryption
15	*k*	Degree of signature

**Table 2 sensors-22-00572-t002:** Simulation setup, configurations, and specifications.

Parameters	Details
Dataset size	100 number of blocks + PHR
Hardware	GPU-enabled system
Software	Ethereum, Hyperledger Fabric
Parameters	Block height, number of blocks, no. transac, no. PHR, delay, signature creation
Performance metric	Efficiency (average percentage of gas, no. packets, no. dead nodes, no. alive nodes),
	security (execution time of policies) and cost (execution time of blocks)
Number of simulations	Number of test performed on single dataset
Number of rounds or transactions	5000

## Data Availability

The data used within the research can be provided by the first author upon request.

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
