# Peer review of "An Industrial IoT-Based Blockchain-Enabled Secure Searchable Encryption Approach for Healthcare Systems Using Neural Network"

_sensors, 2022, doi:10.3390/s22020572_

Round 1

Reviewer 1 Report

This paper implemented the novel comprehensive approach of homomorphic encryption in the digital healthcare system using blockchain technology that provides a secure keyword search facility at the user’s end. The mechanism allows blockchain users to encrypt data at their side and upload to the distributed ledger for record purpose. It provides resistance to active cooperation and replays attacks due to the flexible policy revocation.

However, there are several weaknesses in the paper, which make the paper cannot be fully appreciated. The shortcomings are identified as follows.

  1. The authors may want to rewrite some sentences since they are confusing or have grammar mistakes. For example, in page1 line 5, the sentence has three ‘based’ words, making it hard to figure out what this line is talking about. In page2 line 72, the sentence “Because of ...” has grammar mistake. Similar problem also exists in page6 line 254 and page 9 line 286. The authors may need to check the grammar mistakes carefully.
  2. In page 3-4, there is a copy of the same content from page 2: “In the Internet of Things,..potential attacks aimed.” However, the reference numbers are different. Actually, the section 1.1 could be shorten and the motivation should be clearer.
  3. There are too many unnecessary abbreviations in the article, for example, decentralized applications (Dapps) in line 128, single write(SW), multiple writes (MW), single read(SR) and multiple read(MR) in line 55 and so on. “ABE” in line 97 is used before before being mentioned in 105. “SSA” is used without being mentioned.
  4. In section 1.2, a detailed literature review should not be contribution.
  5. Mistake about figures: line 256, there is no figure below the line, and figure 1 is not matched with the words in line 255-265. Figure 2, 4, 5, 6, 7, 8, 9 have wrong figure title. Figure 5-9 are blurry. In the paper, it happens many times that the mentioned figure does not match the content, such as line 357, line 362, line 379
  6. The equations in page7 need detailed explanation. It is confusing that a bunch of equations are listed in an article, without enough illustration.
  7. The authors fail to cite several past literatures (e.g., [1-3]) highly related to this work, and clearly discuss the differences between them and this paper.

[1] zk-AuthFeed: How to Feed Authenticated Data into Smart Contract with Zero Knowledge. Blockchain 2019.

[2] Incentive Mechanism for Privacy-Aware Data Aggregation in Mobile Crowd Sensing Systems, TON 2018.

[3] Towards fair and efficient task allocation in blockchain-based crowdsourcing. CCF Trans. Netw. 2020.

  1. In section 4, it is not clear whether the pseudo code 1 and 2 are the so called “proposed secure search algorithm”.
  2. In section 4.2-4.6, the author used “we have proposed” many times, but the details are not clear about the proposed algorithm or frameworks, and the mentioned figures does not match the words. Figure2 seems an ordinary neural network, not the neural network designed in this paper.
  3. In page 13, T3 and T4 should be in independent paragraph respectively.
  4. In experiment section. The description of experiment 1 and 2 is not clear enough. For each result figure, it is not clear whether it comes from experiment 1 or 2. Some conclusions cannot be found in the figure, such as line445, line 450, line 459, line 453-458. Conclusion from fig 9 is not given.
  5. In section6, line 489, “will be solved” should be “have been solved”. Future improvements should be placed at the end of the paragraph.

There are some typos in the article, for example:

Line 278, “won” should be “own”, “Our” should be “our”.

Line 305, comma should be added after “moving with”.

Line 347, “its” should be “it is ”, “consist” should be “consists”.

Author Response

Please read the attached response to reviewer 1

Reviewer 2 Report

I have carefully read the article presented. The authors describe a very modern and detailed methodology for systems afferent to the health area. In fact, this argument is very current and heard in many Western countries where the need for security and the demand for protection of personal data are constantly increasing but, systems often face new challenges and risks of an IT and other type. nature. I believe that the authors' proposal is well structured and that the article can be published. However, I believe that the authors in the part of the discussion could mention the fact that greater IT security of healthcare realities also means greater security for patients and therefore becomes part of the clinical risk. For example, this work doi could be cited: 10.1186/s12913-018-3846-7. to explain the efforts being made internationally on this issue.

Author Response

Please read the attached file to reviewer 2

Round 2

Reviewer 1 Report

The authors mentioned in the response that letter that the three papers (i.e., [1-3]) that I suggested in my review have been added in the revised version. However, I do not find them in the revised version. 

[1] zk-AuthFeed: How to Feed Authenticated Data into Smart Contract with Zero Knowledge. Blockchain 2019.

[2] Incentive Mechanism for Privacy-Aware Data Aggregation in Mobile Crowd Sensing Systems, TON 2018.

[3] Towards fair and efficient task allocation in blockchain-based crowdsourcing. CCF Trans. Netw. 2020.

Furthermore, the index of the related work are missing after [29], and several cross references in the paper are not properly showed. Please also correct them in the next version. 

Author Response

 Please read the attached document.

This manuscript is a resubmission of an earlier submission. The following is a list of the peer review reports and author responses from that submission.